# The Role of Nurses Caring for Children Diagnosed with Sickle Cell Anemia and Their Families in a Hospital Setting: A Rapid Review of the Recent Literature

**DOI:** 10.3390/healthcare13040413

**Published:** 2025-02-14

**Authors:** Eduarda Freitas, David Loura, Mariana Inês, Carla Martins, Inês Duarte

**Affiliations:** 1Hospital Dona Estefânia, Local Health Unit of St. José, St. Jacinta Marto, 1169-045 Lisbon, Portugal; davidsloura@gmail.com (D.L.); mariana.ines@ulssjose.min-saude.pt (M.I.); carla.martins2@ulssjose.min-saude.pt (C.M.); ines.duarte@ulssjose.min-saude.pt (I.D.); 2Nursing School of Lisbon (ESEL), Av. Professor Egas Moniz, 1600-096 Lisbon, Portugal

**Keywords:** sickle cell anemia, pediatrics, nursing, hospital

## Abstract

**Background**: Sickle cell anemia (SCA) affects a significant number of children worldwide, for whom the progression of the disease can lead to functional disability-impaired development. Nurses are pivotal in providing holistic care to these children and their families. This review aims to identify recent evidence on the role of nurses in intervening with children with SCA and their families in a hospital setting. **Methods**: A rapid review reported under the PRISMA methodology was carried out in the CINAHL, MEDLINE, and Scopus databases with the expression (sickle cell anemia OR sickle cell disease) AND (child* OR family OR pediatric*) AND (nurs* OR “nursing interventions” OR “pediatric nursing”) AND (hospital*), considering studies between 2019 and 2024, written in English, identifying articles with insights about the role of nurses in this context. Articles other than primary or secondary studies were excluded. Data were analyzed through a rapid qualitative approach. **Results**: Fifty-two studies were identified and seventeen articles were included. The nurse’s role is key and multidisciplinary, focusing on the child and family (care management and therapeutic education), the team (training, and the promotion of safety and quality of care), and the health system (optimizing access to care and promoting adequate resources for its implementation). Such a role is important for short-term clinical problems and to prevent long-term complications. **Conclusions**: Nurses play a central role in empowering families and coordinating multidisciplinary care. Greater investment is needed at a clinical level, through a more effective response to the needs of these patients, and in research, through experimental studies and other designs focused on multidisciplinary interventions.

## 1. Introduction

From a nursing perspective, sickle cell anemia (SCA) presents a complex challenge requiring specialized care to manage its complications and improve patient outcomes. SCA is an autosomal recessive genetic disease that affects millions of people worldwide [1]. This hemoglobinopathy is caused by a mutation of a single nucleotide in the beta chain of hemoglobin, which leads to the existence of an altered structure called hemoglobin S, which, unlike normal hemoglobin, undergoes polymerization when exposed to challenging conditions, causing changes in the shape, rigidity, and function of erythrocytes [2]. Despite being a monogenic disease, it has multisystemic expression and is therefore characterized by a vast multiplicity of clinical complications. These can be both acute and/or chronic and reflect the complex pathophysiology of vaso-occlusion and hemolysis [3].

SCA is particularly prevalent in populations of African origin but is also found in groups from the Mediterranean region, the Middle East, and India. It is estimated that 300,000 children are born each year with SCA, with the majority of those affected living in sub-Saharan Africa, India, the Mediterranean, and the Middle East [4]. SCA represents a significant challenge for public health, being responsible for high infant morbidity and mortality. In low-income countries, up to 90% of children with the disease may die before the age of five due to complications such as infections, vaso-occlusive crises, and stroke, if there is not early diagnosis and adequate treatment [5].

In the hospital setting, nurses play a crucial role in the care of children with SCA, addressing not only the physical manifestations of the disease but also its psychological and social impact. In a pediatric context, SCA represents a significant challenge due to its premature presentation and the complications that arise throughout development, which require early intervention and multidisciplinary monitoring, with the aim of improving the prognosis and quality of life of these children and their families [6]. These children are at increased risk of developing acute and chronic complications, such as infections, vaso-occlusive crises, liver crises, splenic sequestration, anemia, priapism, acute chest syndrome, strokes, and delayed growth and development [7].

In order to optimize the diagnostic process, the World Health Organization has overseen and encouraged the existence of screening programs in general, such as neonatal screening [8]. In 2023, the WHO’s Department of Maternal, Child and Adolescent Health and Aging recognized the need to promote universal neonatal screening programs to diagnose priority health conditions, including hemoglobinopathies, which include SCA [9]. This can be important so that a diagnosis can be made as early as possible and, together with the advances in treatment promoted in recent years, namely, antibiotic prophylaxis, blood transfusions, and the use of hydroxycarbamide, an effective improvement in the quality of life of affected children can be promoted [10]. However, there is still significant variability in response to treatment, and ongoing research into new therapeutic approaches and long-term care is needed [11].

Given the need for advanced therapeutic management and to prevent complications in the context of acute SCA, children with this pathology are often admitted to the hospital [12]. The period of pediatric hospitalization is a time of disruption in the daily dynamics of children and their families, which can have a negative psycho-emotional impact [13]. The family suffers from the hospitalization process and can experience uncertainty, insecurity, feelings of powerlessness, among other things, which can have a direct impact on the child [14]. Nurses are fundamental in mitigating these effects, providing not only medical care but also emotional and psychological support to both the child and their family. In order to transform hospitalization into a less traumatic process, studies have shown that nursing must act with a variety of care actions, one of the greatest allies being the process of training, therapeutic education, and empowerment of the child and family [15].

Roy’s adaptation model is an epistemological approach that can be applied to the experience of SCA in a pediatric context, seeking to frame and explain the adaptive responses of children and their families, considering the physical, psychological, and social stress factors associated with hospitalization and dealing with the disease. According to Roy, people are adaptive systems that continually interact with the environment and are challenged by stimuli that affect their adaptive capacities [16].

From clinical experience, it is clear that the role of nurses is fundamental in caring for children with SCA is important, since this is a condition that requires frequent and careful interventions due to the complexity and severity of the disease and its complications. Nurses play a pivotal role in the care of hospitalized children with SCA, acting as essential mediators between the child, family, and healthcare team. Beyond managing physical symptoms and treatment adherence, nursing care encompasses emotional and psychological support, family education, and strategies to promote self-care [17]. Given the multidimensional impact of SCA—ranging from acute complications to social and emotional challenges such as stigmatization, anxiety, and depression [18]—nurses employ a holistic and humanized approach to minimize suffering, prevent complications, and enhance the child’s overall well-being [19]. Their frontline presence enables early complication detection, therapeutic interventions, and continuous education, which are fundamental to improving quality-of-life and long-term health outcomes for children with SCD [20]. The ongoing dissemination of information and development of targeted nursing strategies are essential to advancing care for this vulnerable population.

Considering the growing body of evidence already produced in this area and the scarcity of secondary studies completed or underway to synthesize it, as evidenced by an exploratory search carried out by researchers in the OSF, PROSPERO, MEDLINE, Cochrane Database of Systematic Reviews, and JBI Evidence Synthesis databases, the motivation for this study is associated with the need to synthesize the most recent knowledge in the literature on the topic. The option for a rapid review was made due to the shattered rhythm of incidence of this disease compared to scientific production to guide health interventions at the field, especially regarding nurses, generating a clear need to identify studies that are potentially transferable to clinical practice. This approach allows for a faster and more precise response to the emerging challenges in healthcare.

This review will try to describe key evidence-based practices that enhance nursing care for children with sickle cell disease. By integrating scientific research, clinical expertise, and patient needs, it will be possible to create effective strategies for clinical practice in this field and to identify knowledge gaps to guide future research. The aim of this rapid review is to identify recent evidence on the role of nurses in interventions with children with SCA and their families in a hospital setting.

## 2. Materials and Methods

The decision to carry out a rapid literature review stems from the need to inform clinical practice in this area about the latest literature with transferable potential to enable evidence-based practice. Thus, this rapid review was conducted taking into account the approach preconized in the literature and by following the Cochrane rapid review methods guidance for rapid reviews of effectiveness [21]. In the absence of published reporting guidelines on rapid reviews and in line with Cochrane’s suggestion, the PRISMA methodology was used to guide the presentation of this review process. The review was registered on the OSF platform and is accessible via the following link https://doi.org/10.17605/OSF.IO/3JTPD (accessed on 29 November 2024).

### 2.1. Eligibility Criteria

In line with the defined objective, the following research question was defined in the PCC format: what recent evidence is available about nursing interventions (P) for children with SCA and their families (C) in a hospital setting (C)?

The use of the PCC format, which is normally used for scoping reviews, was adopted instead of the PICO format due to the nature of the research question. The PCC framework aims to structure the research question in population, concept, and context, which is suitable for this rapid review given that the aim is to identify recent evidence on the role of nurses in interventions with children with SCA and their families in a hospital setting, The PICO framework is more suitable for reviews where there is a need to identify the outcome of an intervention while comparing it to other factors in a certain population. Since this rapid review does not intend to introduce comparators or outcomes a priori, the PCC framework was the chosen approach for constructing the research question.

In terms of participants, the review considered all studies that included children between the ages of 0 and 18 with a diagnosis of SCA, as well as studies that included family members of these children, regardless of the degree of kinship or type of family. The choice of defining the family as an inclusion criterion is linked to its importance in the context of hospitalization and in the management of a child’s chronic illness, and the search for nursing interventions aimed at this population is relevant to the review question. Studies that included people diagnosed with other hemoglobinopathies or at other ages were also considered, if it was possible to extract results and/or conclusions related solely and exclusively to the review population. On the other hand, studies that included healthy children or those diagnosed with other chronic diseases were excluded, as were studies that exclusively included the adult population.

For the concept, the definition drawn up by the International Council of Nurses was used, in which a nursing intervention is an action taken in response to a nursing diagnosis to produce a nursing outcome [22]. Thus, all studies that mentioned nursing care or interventions were included, regardless of the diagnoses or associated outcomes, as well as the taxonomy used and the level of specialization of the nurses responsible for their implementation. Studies that mentioned nursing interventions as part of multidisciplinary programs, or that identified multidisciplinary interventions with the participation of nurses, or with implications for nursing care, were also considered. Studies that addressed interventions without nurse involvement or without implications for nursing and that were exclusively conducted by other professionals, as well as interventions not directly related to the provision of direct care, were excluded.

In this context, all studies focusing on the process of clinical follow-up and care provision in a hospital setting for children diagnosed with SCA and their families were included, regardless of where this follow-up could take place (patient, emergency department, outpatient, day hospital, operating room, among others). Studies that addressed non-pediatric clinical practice environments, as well as care processes in prehospital and community settings, were excluded. No sociodemographic restrictions were applied.

In regard to the type of evidence, primary studies of a quantitative, qualitative, or mixed nature, and published reviews of the literature (narrative or systematic) were considered. Protocols, letters to the editor, perspectives or discussion articles, editorials, columns, commentaries, case studies, and book reviews were excluded. The reason for this exclusion is that these works do not present a sufficient level of evidence to transfer knowledge to the clinical context in future studies that address the results of this review.

### 2.2. Search Strategy

In line with Cochrane’s recommendations, the definition of the research strategy included two phases. Initially, an exploratory search was carried out in databases appropriate for the subject under research (CINAHL, MEDLINE, MedicLatina, Scopus, Web of Science, and Academic Search Complete), with the aim of gaining a better understanding of how the concepts are described in the literature. This step was essential for identifying the most appropriate terms to describe the concepts inherent in the research question, which in turn was essential for designing a search expression with the greatest possible specificity and sensitivity. In this sense, a test search expression was developed during this process and applied to the databases mentioned above. It became clear that this area is a field of heuristic availability, and not all databases returned results.

The second phase corresponded to the search itself, which was rerun on the Scopus, CINAHL, and MEDLINE databases. The choice of these databases was linked to their multidimensional nature and to higher probability of returning results, given that these were the databases with more articles available in the previous exploratory search.

Thus, the search was carried out in the Scopus, CINAHL, and MEDLINE databases, using the final search expression in natural language:

(sickle cell anemia OR sickle cell disease) AND (child* OR family OR pediatric*) AND (nurs* OR “nursing interventions” OR “pediatric nursing”) AND (hospital*).

This expression was applied in September 2024 and it was designed after group discussion (triple-checking) considering the results of the exploratory research, after it was tested and refined. The option for natural language terms was made given the higher number of results compared with indexed terms. Filters were used to delimit the search, such as the English language and the year of publication between 2019 and 2024. The time restriction is associated with the need to identify recent evidence on the subject.

### 2.3. Selection of Evidence

Once the records from the search were obtained, they were extracted and uploaded to the bibliographic management support software Zotero (version 6.0.36) and the bibliographic review support system Rayyan [23], where the eligibility assessment process began. There was no piloting or agreement between reviewers, as the process was always conducted by two independent reviewers at all stages. Initially, the articles were analyzed for their agreement with the inclusion criteria by title and abstract. After this process, the reviewers proceeded to the full-text review, where they assessed the relevance of the results and conclusions of each study with the aim of answering the defined review question. Conflicts in decisions regarding the selection of articles were resolved by consensus, and there was no need to call a third reviewer. The reasons for excluding the articles were categorized according to their incompatibility with the defined inclusion criteria and summarized, together with the data on the selection process, in a PRISMA flow diagram (Figure 1).

### 2.4. Data Extraction

After the full texts of the articles were analyzed, the most relevant data were extracted via a previously constructed tool, the design of which was approved by all members of the review team (including fields such as authors, year, country, methodology, and main results). The data extraction process was led by one of the reviewers, and another reviewer was responsible for checking and critically analyzing the extracted information, as recommended by the Cochrane Library, with an inter-reviewer reliability score of 98%. The data included in the main results was then reorganized in order to align with the research question.

Regarding quality appraisal, and given the diverse nature of results, several tools were used. The tool from Hawker et al. [25] was used to evaluate the quality of primary studies. This tool includes areas of analysis to evaluate each article by asking questions such as, “Does the abstract and title provide a clear Does it provide a clear description of the study?” or “Is there a good basis and clear explanation of the aim of the study?”. These questions were scored from 1: very poor to 4: good, and a compliance rate was calculated (obtained score/maximum score).

Since the previous tool is only applicable to primary studies, the JBI recommendations were adopted for the appraisal process of narrative and systematic reviews [26,27,28], consisting of questions on the nature of the studies with the answers “yes”, “no”, and “not clear”. In order to better assess the quality of the studies according to the JBI quality standards, the authors calculated a compliance index corresponding to the percentage of responses meeting these standards (number of “yes” responses/total number of questions). The authors analyzed the articles and debated until they could reach a consensus. A general compliance rate was also calculated. No studies were excluded based on the results of this appraisal process.

### 2.5. Data Analysis and Presentation

The data from this review is presented in a structured and schematic way, with the aim of broadening the reader’s understanding of the results of the review. A PRISMA flow diagram showing the process of identifying, assessing eligibility, and including the articles is also shown in Figure 1 in Section 2. The data were analyzed by an inductive approach under rapid qualitative research methodology [29], i.e., a categorization matrix was drawn up on the basis of the results and in line with the review question.

## 3. Results

The identification of the studies and their selection process is summarized in the PRISMA flowchart in Figure 1. Among the eighty-two articles identified, and after removing thirty duplicates, thirty-four articles were removed by screening their titles and abstracts and one was removed by full text, leaving seventeen studies to be included in this review (Table 1). The decision to exclude one article by full text was made due to the fact that, although nurses were involved in the study, its focus was not their intervention and, therefore, it did not have a meaningful contribution to the present review.

The majority were primary (n = 14, 82%), quantitative (n = 8, 57%), or qualitative (n = 6, 43%) studies, whereas the remaining studies were secondary literature reviews (n = 3, 18%). With respect to the year of publication, most of the studies were published between 2023 and 2024 (n = 8, 47%), and there were also publications in other years with less expression. The articles selected provide a general overview of SCA worldwide, as they were carried out in several countries, mainly in the United States of America (n = 8, 47%), Canada (n = 2, 12%), and Brazil (n = 2, 12%). A quality appraisal of the included articles was performed (Appendix A) and a compliance rate of 83% is reported. In these articles, the nurse’s intervention was described as multidimensional, focusing on the child and family, the team, and the health system, as illustrated in Figure 2, which will be explored in more detail in the following paragraphs.

**Figure 2 healthcare-13-00413-f002:**
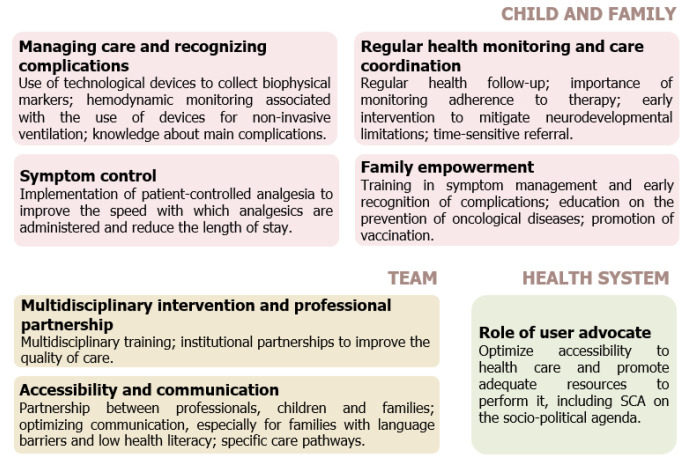
Schematic illustration of the main results of the review. Regarding child and family, managing care and recognizing complications are mentioned by references [30,31,32,33,34,35,36,37,38,39], regular health monitoring and care coordination are mentioned by references [31,35,40,41], symptom control is mentioned by references [32,33,34,35,36,37,38,39,42,43] and family empowerment is mentioned by references [32,35,36,38,42,44,45]. Concerning the team, multidisciplinary intervention and professional partnership are mentioned by references [35,36,37,38,40,45] and accessibility and communication are mentioned by references [31,33,34,35,41,46]. In the health system category, the role of user advocate is mentioned by references [32,35,38,40,44,46].

**Table 1 healthcare-13-00413-t001:** Summary of selected articles.

Authors	Year	Country	Aim	Methodology	Participants	Measures/Outcomes	**Main Results**
**Ajayi et al.** [30]	2021	USA	To evaluate the ability of wrist-wearable technology to collect physiological data from children with serious illnesses	Type of study Primary study—Quantitative Design/Methods Observational and prospective design	Twelve pediatric patients with diagnoses of cancer and sickle cell disease admitted to the hospital for acute-on-chronic pain taking opioid medications	Heart rate, respiration rate, temperature, patient-reported physical and emotional symptoms.	The collection of biophysical markers using technological devices is a reliable way of predicting changes in the vital signs of children with SCA, particularly with regard to the presence of pain, and is a less invasive alternative to the conventional method. A main limitation of this study is the overall small sample size.
**Alghubaishi and Aldahmashi** [32]	2024	Saudi Arabia	To address the impact of supportive care interventions on QoL in pediatric hematologic disorders	Type of study Systematic review with meta-analysis	Patients with hematologic disorders submitted to supportive care interventions	Quality of life (physical functioning, emotional functioning, social functioning, and school functioning) and patient-reported satisfaction with the supportive care	Nurses are important players in providing care that promotes quality of life for children with SCA. Their intervention should be multidisciplinary, establishing a bridge with other professionals for artistic and school activities, social support, and adapted physical exercise. The most significant impact of the nurses’ intervention is in terms of empowering the parents and close family, as well as the child, promoting proper symptom management and early recognition of complications and warning signs. Main limitations are related to the study’s heterogeneity and small number of studies with statistical data for the meta-analysis.
**Arbitre et al.** [34]	2021	Canada	To review the use of patient-controlled analgesia (PCA) in sickle cell disease (SCD) for pediatric patients with vaso-occlusive crises (VOC) and compare its effect related to their time of implementation	Type of study Primary study—Quantitative Design/Methods Observational and retrospective design	A total of 87 pediatric patients with SCD treated with early PCA (n = 63) and late PCA (n = 24)	Patient relief and length of stay	In the context of vaso-occlusive crises, the implementation of patient-controlled analgesia solutions provides faster administration of analgesia and shorter length of stay. Main limitations concern the low number of patients receiving PCA during the study period and the specifics of using a retrospective design.
**Aurora et al.** [44]	2023	USA	To evaluate the effectiveness of a vaccine strategy bundle to increase human papillomavirus (HPV) vaccine initiation and completion in a specialty clinic setting	Type of study Primary study—Quantitative Design/Methods Implementation framework	A total of 479 adolescents with SCA eligible for HPV vaccination	HPV vaccination series completion	Educating children and adolescents with SCA and their families about the prevention of oncological diseases is very important, given the higher risk these children have of developing these diseases. HPV vaccination is an important strategy in which nurses can play a very important role. Main limitations regard the use of retrospective data and transferring these findings into resource-limited settings.
**Borges et al.** [42]	2020	Brazil	To know the vision of the child and adolescent with SCA about the healthcare they receive.	Type of study Primary study—Qualitative Design/Methods Talking map technique, data analyzed through content analysis.	Five children and two adolescents with SCA	Disease and treatment-based knowledge, accessibility to healthcare and hospital care.	Living with SCA is seen by children and adolescents as a multidimensional process in which they adquire meaningful knowledge and practices, but also as a burden given the recurrent hospitalizations, inefficent care transitions, and low integration of care. For these children, professionals shall be able to value their pain and act efficently to reduce their suffering and humanize the hospital environement, which has a stressful potential for them. Therapeutic play has a major importance to minimize the negative impact of a hospital admission in a child with SCA. Adolescents should have adequate treatment given the specificity of this age group. Nurses have a pivotal importance regarding emotional support, clear communication, and education tailored to the child’s understanding. Limitations relate to the specificity of the setting where the study was conducted (university hospital).
**Cohen et al.** [33]	2024	USA	To identify perceived benefits, harms, facilitators, and barriers to use of supportive non-invasive ventilation for acute chest syndrome prevention for hospitalized children with sickle cell disease (SNAP)	Type of study Primary study—Qualitative Design/Methods Semi-structured key informant interviews to support implementation.	A total of 34 participants, including doctors, nurses, respiratory therapists, child life specialists, psychologists, and young people with SCA and their parents	Benefits, harms, facilitators, and barriers to use of SNAP	The implementation of noninvasive ventilation is seen as an effective and appropriate therapeutic strategy for hospitalized children with SCA, as it can prevent acute chest syndrome and respiratory distress. The biggest challenges and limitations, in which nurses can play a decisive role, relates to improving the experience and acceptance of the child and family, as well as the hemodynamic surveillance and monitoring associated with the use of these devices.
**Ghafuri et al.** [40]	2021	Nigeria	To describe the opportunities and challenges in setting up and conducting a clinical trial on hydroxyurea treatment in a low resource setting.	Type of study Primary study—Quantitative Design/Methods Observational descriptive and prospective design	A total of 679 children with SCA	Priorities for capacity building on stroke prevention in children with SCA and professionals caring for them	Multidisciplinary training for the teams responsible for caring for children with SCA and their families is extremely important, especially with regard to stroke prevention. No limitations were described.
**Heitzer et al.** [46]	2024	USA	To identify determinants influencing the utilization of early intervention services among young children with sickle cell disease (SCD) based on perspectives from medical and early intervention providers	Type of study Primary study—Qualitative Design/Methods Telephonic interviews analyzed with inductive analysis	Twenty health professionals associated with early intervention and medical intervention	Awareness, access and communication as determinants for early intervention services use by children with SCD	The use of early intervention teams by children with SCA is extremely important to mitigate the neurodevelopmental limitations they experience. The barriers to intervention are related to the lack of awareness, difficult access, and limited communication between professionals and families. Main limitations are associated with the conduction of interviews virtually, previous work relationships between researchers and participants, and limited data collection on demographic traits.
**Houwing et al.** [35]	2021	Netherlands	To map best practices and lessons learnt in order to attain more optimal healthcare accessibility for pediatric patients with sickle cell disease and their families.	Type of study Primary study—Qualitative Design/Methods Semi-structured interviews analyzed through thematic analysis	Twenty-two health professionals from five university hospitals, including pediatric hematologists, nurses and other professionals	Six themes emerged associated with best practices on topics related to the improvement of healthcare accessibility for this population: cutting invisible costs, reducing the number of hospital visits, specialized and shared care, optimizing communication, building digital connections, and patient-centered support	Access to adequate healthcare for children with SCA is hampered by socioeconomic and health system-related barriers. Possible solutions to these problems are: full reimbursement of the costs associated with managing SCA by children and families; aggregation of appointment scheduling and clinical procedures to avoid unnecessary trips to the hospital; development of a partnership between professionals, children and families; optimization of verbal and written communication, with special attention to families with language barriers and low health literacy; improvement of eHealth services to make them more accessible to people with low digital literacy; putting SCA on the political agenda, making the challenges of people suffering from this disease visible. The main limitation relates to the low sie of the sample.
**Kingsley** [36]	2020	USA	To increase multidisciplinary pain management referrals for youth with SCD identified to be at risk for chronic pain	Type of study Primary study—Quantitative Design/Methods Implementation design	A total of 111 children greater than 2 years of age and less than 21 years of age with laboratory confirmed SCD	Pain management referrals	Treating pain in children with SCA and preventing chronic pain is an aim that can be aligned with multidisciplinary pain management referrals. Nurses are present in the whole spectrum of healthcare in which these children and families are present. Providing comprehensive education about pain and conducting rigorous and multidimensional pain assessment, treatment, and re-assessment is the duty of nurses aiming to achieve confort for these patients. Working with multidisciplinary teams to this end draws an opportunity to achieve long-term positive outcomes. The absence of a valid and reliable screening tool for the aim of this project is noted as the main limitation.
**Martin et al.** [37]	2020	USA	To evaluate ED provider awareness of and comfort with current pain management guidelines for sickle cell vaso-occlusive crises.	Type of study Primary study—Quantitative Design/Methods Observational and cross-sectional design	A total of 52 health professionals working in an emergency department of a pediatric hospital (attendings, resident trainees and nurses)	Knowledge and confort about vaso-occlusive management guidelines	Nurses feel confident in managing vaso-occlusive crises in children with SCA, although less than 10% knew the recommended time for the initial administration of analgesia in a pediatric emergency setting. There are still challenges associated with assessing vital signs related to the presence of pain and monitoring pain with appropriate scales. Small sample sizes and similar training between providers are the main limitations.
**Razeq et al.** [43]	2024	Jordan	To examine nurses’ attitudes towards caring for children with sickle cell disease (SCD) and SCD pain management in those with vaso-occlusive pain	Type of study Primary study—Quantitative	A total of 298 nurses caring for pediatric patients with SCA.	Stigmas and stereotypes, attitudes toward pain management, attitudes concerning opioid pain management, and hesitancy in administering opioid-based analgesia	Pain management is a priority in the treatment of vaso-occlusive crises in children with SCA. It is therefore extremely important that the entire team is trained to act in these situations, which can be optimized by attending specialized training in pain management and in the administration of opioids. There are barriers to optimum pain management in a hospital setting, particularly those related to the overload and insufficient time to properly monitor and supervise the pain of children with SCA. Main limitations were related to the pre-existing knowledge of nurses about comorbidities or specificities that may affect pain management.
**Reece-Mills et al.** [38]	2023	Canada	To outline the core program components and identifies major successes and challenges of the SickKids-Caribbean Initiative	Type of study Primary study—Qualitative Design/Methods Implementation design	Health professionals involved in the training and care of children with cancer and hematological diseases in the partner countries (Barbados, Bahamas, Jamaica, Saint Lucia, Saint Vincent and the Grenadines, and Trinidad and Tobago)	Use of specialized knowledge and skills in patient management, use of data to improve clinical care and decision-making, knowledge translation and dissemination of evidence-based practice, enhanced regional capacity for care, clinical expertise for diagnosis and management of these patients, equitable access to care, and the quality of life of children.	The role of the health team in optimally managing the needs of children with SCA begins before they go to hospital, and institutional partnerships are of unique importance for improving the quality of care by sharing experiences and putting this issue on the sociopolitical agenda. An interinstitutional partnership is reported, in which the following activities related to the nursing area of competence were carried out: human resources planning and specialized training; patient advocacy initiatives with the involvement of civil society; integrated and sustainable management. Limitations regard site-specific differences and local consideration to multiprofessional training.
**Telfer** [31]	2019	UK	To discuss the presentation and management of acute sickle crises, highlighting which aspects of diagnosis and management can be undertaken in the community and which require urgent referral to hospital	Type of study Narrative review Design/Methods Clinical report	Pediatric patients with SCD	Acute presentations of SCD crises, guidance for clinical practice.	Children with SCA have acute crises that require advanced and early treatment. Nurses shall be aware of different acute presentations to be able to provide effective and safe care, referring to the hospital with the right timing to maximize outcomes and prevent complications. Dactylitis, acute splenic sequestration, acute painful crises, acute chest syndrome, acute stroke, abdominal crises, acute anemic episodes, priapism, and acute visual problems. Specific care pathways for these children should be created to better manage these complications and act efficiently on pain with multidisciplinary monitoring and counseling. No limitations are described.
**Shaner et al.** [39]	2021	USA	To evaluate the impact of telehealth dosing adjustments on hydroxyurea laboratory and clinical response as compared with university-based patients	Type of study Primary study—Quantitative Design/Methods Observational and retrospective design	A total of 107 patients treated at university centers and 65 patients treated at outpatient clinics	Number of clinic and acute care visits for one year, mean complete blood count, and fetal hemoglobin (HbF) levels	Monitoring adherence to therapy is an activity with a high impact on the clinical stabilization of children with SCA, and the use of telehealth tools to monitor these patients is apparently an option with effective success, particularly in those living in rural areas. Difficulties in monitoring adherence to treatment and recall bias are the main limitatons.
**Teixeira et al.** [45]	2023	Brazil	To know the perception of nurses about the child with sickle cell disease	Type of study Primary study—Qualitative Design/Methods Semi-structured interviews, convergence groups and participant observation, analyzed through convergent-care research.	A total of 12 nurses from the pediatric emergency department of a public pediatric hospital	Nurses’ knowledge about SCD and nurses’ knowledge about the child with SCD	Pain management is one of the aspects most valued by nurses in terms of the needs of children with SCA, while there are still gaps in terms of other aspects of the pathology that can lead to therapeutic needs. Training is an important area of action to improve this knowledge profile. The main limitation is related to the site-specific nature of the study.
**Yates et al.** [41]	2024	USA	Provide an overview focused on the practical management of children and adolescents with SCA and the complications particularly relevant to pediatric health professionals.	Type of study Narrative review Design/Methods Clinical report design	Children and adolescentes with SCD	Pathophysiology, diagnosis, complications, chronic manifestations, supportive care, health maintenance, genetic education, and counseling, transition and health supervision	Providing comprehensive care is a time-consuming effort and includes ongoing patient and family education, periodic comprehensive assessments and other disease-specific health maintenance services, nursing support, psychosocial care, and genetic counseling. No limitations are described.

Underlined designations “Type of study” and “Design/Methods” serve the purpose of consistency regarding the categorization of the studies’ methodologies, aiming to quickly inform the readers about these two features.

### 3.1. Children and Family

#### 3.1.1. Managing Care and Recognizing Complications

Care management and early recognition of complications are cornerstones in the care of children with SCA. The role of nurses is central to ensuring their safety and well-being. This role involves the use of technological devices for continuous monitoring of biophysical markers such as heart rate and oxygen saturation; electrodermal activity; movement-based activity; and temperature [30]. This use has proven to be a useful tool in the continuous monitoring of pediatric patients’ health, being useful for identifying changes early in life and thus helping to make quick and accurate clinical decisions and time-sensitive referrals in community settings [30,31]. In this way, the data are essential for identifying early signs of decompensation that may precede vaso-occlusive crises, acute chest syndromes, and other complications common in these children [31,32]. Hemodynamic monitoring allows close monitoring of cardiovascular stability, which is essential during acute pain crises and episodes of hypoxemia [33].

#### 3.1.2. Regular Health Monitoring and Coordination of Care

Regular health monitoring and coordination of care constitute another central axis in the care of children with SCA, with nurses being identified as those responsible for ensuring continuous and integrated monitoring [40]. Some nursing interventions include periodic health assessments, such as physical examinations, laboratory tests, and checking for symptoms, enabling early identification of possible complications [35]. According to Yates et al. [41], regular consultations with a multidisciplinary team help with the early detection of complications.

#### 3.1.3. Symptom Control

Effective symptom management in children with SCA is an essential component in the treatment of this disease, especially regarding pain control, which is one of the most debilitating and common manifestations [36], especially in the case of vaso-occlusive crises [37]. Nursing plays a central role in this process by providing a holistic and individualized approach. However, in addition to pain management, which encompasses the dynamics of continuous assessment of this phenomenon and the application of individualized treatment strategies, including the use of opioids [43], there are other interventions described by the evidence to manage associated symptoms and possible complications, such as blood transfusions, antibiotic therapy [42], and the use of noninvasive ventilation in the treatment of acute chest syndrome [33]. Appropriate verbal communication and the use of mobile applications to optimize access to information and improve engagement in care facilitate the implementation of these interventions (Houwing, 2019), as does psychosocial support, which includes fostering visits from extended family and encouraging play [35,42]. These measures are important not only for managing the child’s symptoms in the immediate aftermath, but also for minimizing the emotional impact of hospitalization and improving the child’s well-being.

Conerning pain management, Arbitre et al. [34] studied the implementation of patient-controlled analgesia (PCA) in children’s hospitals and reported that this strategy improved the speed with which analgesics were administered and contributed to a reduction in the length of stay. This method allows patients to adjust the dosage within safe limits, minimizing the risks of overdose or inadequate control, reducing the wait for the medication to be administered and increasing the patient’s autonomy in controlling their pain [32,34]. Compared with those who received traditional analgesia, children who received PCA had a faster response to pain treatment, less need for continuous medication adjustments, and a lower incidence of side effects [34].

The implementation of PCA has positive results not only in pain control but also in the efficiency of hospital care. Thus, supportive interventions, such as PCA, benefit more effective management of hospital resources by reducing hospitalization time, since the administration of more effective analgesics allows recovery in a shorter space of time, preventing complications, and improving the child’s general well-being [32]. Improved pain control and shorter hospital stays have a direct effect on the quality of life of children with SCA and their families. Access to high-quality interventions that improve the experience of pediatric patients, highlighting strategies such as PCA and therapeutic play, is necessary to provide patient-centered care, reduce the stress associated with hospitalizations [35,42], and improve long-term health outcomes [32,34]. This will help professionals to better understand the suffering of these children, giving them a more pleasant experience during a hospital stay [42].

#### 3.1.4. Family Empowerment

Empowering children with SCA and their families is essential for the effective management of the disease, especially in the early recognition of complications and the adoption of preventive measures. According to Houwing et al. [35], the continuous education of families in partnership with health professionals has improved the immediate response to vaso-occlusive crises and other acute complications. Recognizing warning signs such as severe pain and fever allows early intervention and reduces the risk of serious complications such as acute chest syndrome and splenic sequestration [41]. The existence of targeted educational programs, adjusted to the level of understanding of each child, implemented in partnership with health institutions, promotes greater knowledge about the risk of complications and encourages the active search for preventive care [36,38,42]. In this way, family empowerment enables continuous monitoring and improves adherence to the therapeutic regimen.

On the other hand, promoting vaccination is also a preventive measure for children with SCA, as they are more susceptible to serious infections. Various strategies, including educational campaigns and vaccination reminders, significantly increase immunization rates in adolescents with SCA [44]. Nurses therefore play a key role in educating and promoting vaccination-seeking behavior, providing information on the importance of having an up-to-date vaccination schedule. This educational support helps families understand the risk of infection and adhere to immunization recommendations, thus ensuring adequate protection [45]. Consequently, the active involvement of families in managing healthcare and preventing complications leads to a better quality of life for these children and their families [32]. This training enables caregivers to quickly recognize warning signs and take precautionary measures to prevent symptoms from worsening.

### 3.2. Team

#### 3.2.1. Multidisciplinary Intervention and Professional Partnerships

The effective management of SCA in pediatrics requires an integrated, multidisciplinary approach that involves close collaboration between different health professionals, as well as efficient communication with children and their families. This strategy is key to addressing the complexity of the disease and providing child- and family-centered care. A multidisciplinary intervention is essential to ensure a holistic and effective approach for children with SCA. The involvement of doctors, nurses, psychologists, social workers, and other specialists makes it possible to meet children’s diverse needs, from symptom control to emotional and psychosocial support [35]. A strategy to better translate this need to clinical settings is the creation of multidisciplinary pain management teams, developing their work through referrals and working for rigorous assessment, treatment, and continuing follow-up of the child’s pain [36].

For a multidisciplinary approach, institutional partnerships are an effective strategy, resulting in significant improvements in the quality of care provided to children with hematological diseases, including SCA. These institutional collaborations allow for the exchange of knowledge and access to additional resources [38].

In addition, Ghafuri et al. [40] and Martin et al. [37] highlighted the importance of training teams that care for these children. Empowering healthcare professionals in resource-limited settings improves care in vaso-occlusive crises and stroke prevention, which are common conditions in patients with SCA. This type of intervention is essential to ensure that health teams can respond effectively to the needs of children with this disease.

#### 3.2.2. Accessibility and Communication

Access to healthcare and the optimization of communication are critical pillars in the management of chronic diseases. According to Heitzer et al. [46], effective communication between healthcare professionals, patients, and their families is key to promoting better adherence to treatment and ensuring a higher quality of care. The partnership between healthcare professionals and families allows for a better understanding of patients’ individual needs and facilitates the development of care plans to reduce suffering [42].

An important challenge is the language barrier and low health literacy, as they can make it difficult for families to understand care plans [35]. Initiatives that address these barriers include making educational materials available in several languages and using visual aids to make treatment instructions easier to understand. The use of telemedicine technology can also help overcome these barriers, especially in areas where access to primary healthcare is difficult [35].

Specific care pathways can also be a positive initiative to provide more dedicated care to these patients, as they have particular needs and susceptibility [31]. Therefore, an integrated approach, with institutional partnerships and a focus on accessibility and communication, is important for improving the care of children with SCA. The combination of continuous training for professionals and the empowerment of families not only improves disease control but also strengthens the support network around the child, contributing to a better quality of life and positive clinical outcomes [41].

### 3.3. Health System

The nurse’s role in advocating for the child and family is key to improving accessibility to healthcare, especially for children with complex chronic diseases such as SCA. Although all health professionals play a role in health advocacy, nurses are especially important, as they act as intermediaries between patients and the system to ensure that their needs are met effectively and equitably [35].

This way, integrating SCA into public health policies and the sociopolitical agenda is a fundamental strategy to guarantee the continuity and improvement of the services provided [38]. Cooperation between different health services and public initiatives can increase access to healthcare and consequently reduce inequalities, particularly in regions where access to specialized healthcare is limited. Studies have shown that partnerships between different public health institutions and policy initiatives have resulted in significant improvements in the quality and accessibility of care for children with hematological diseases [35].

Advocacy also plays a vital role in health education, promoting knowledge of the disease and encouraging vaccination policies to prevent associated complications. For example, a multicomponent strategy adopted to increase HPV vaccination rates in adolescents with SCA has proven to be an effective approach [44].

Finally, optimizing accessibility to care and advocating for the needs of children and their families through specific public policies and constant advocacy ensures that they receive quality care. The continued inclusion of SCA in the sociopolitical agenda is crucial to ensure adequate resources and the maintenance of specialized services [32,46].

## 4. Discussion

SCA is one of the most common genetic diseases worldwide, presenting considerable challenges in the provision of care, especially in pediatric patients [47]. This study sought to identify recent evidence on the role of nurses in intervening with children with SCA and their families in a hospital setting. In this sense, the results met the outlined objective.

With respect to scientific evidence, one of the characteristics to highlight is the recent year of publication of all the articles, with five studies from 2024, three from 2023, five from 2021, and four between 2019 and 2020, which indicates a growing trend of publications over the years in this sample. The studies selected were carried out in eight different countries, illustrating the multiculturalism and diversity of the populations affected by SCA. In addition to the relevance of multiculturalism in rapid literature reviews, the geographical variation in this study is even more pertinent because SCA is a hereditary disease that occurs due to mutations in the hemoglobin gene. Studying several countries allows better understandability about the different manifestations of the disease and the treatment strategies adopted in genetically diverse issues. This is crucial, as the genetic mutation in each region can influence the manifestations of the disease and the response to treatments [35].

Another aspect to mention is that the approach to SCA in children varies widely according to available resources and cultural practices. For example, in developed countries such as the United States and Canada, there seems to be greater access to hydroxyurea treatments and blood transfusions. In developing countries, such as Nigeria and Brazil, limited access to healthcare, less availability of drugs, and difficulties in adhering to drug regimens are major challenges. These differences highlight the need to adapt to each population [32,40]. Thus, research varies according to country and can focus more on therapeutic advances and child-centered care models (United States and Canada), accessibility challenges, health education, and treatment adherence strategies (Brazil), health education programs and prevention strategies (Nigeria), cultural and religious practices that influence the treatment of the disease, and the impact of educational interventions on the early recognition of complications (Saudi Arabia and Jordan), and immigrants in countries, thus offering insights into the integration of healthcare in minority groups (The Netherlands) [35]. Therefore, the fact that the included studies were carried out in different locations and socioeconomic contexts increases the relevance and applicability of the results, offering a comprehensive view of how different health systems care for children with SCA.

In addition, it is also clear that this topic is discussed by various health disciplines given the multiplicity of scientific backgrounds of the authors of the selected articles, namely, nursing, pediatrics, hematology, public health, neurology, and digital health. This diversity of fields is essential for addressing the complexity of SCA, especially in pediatric settings, where caring for these children requires a broad, coordinated, and innovative approach.

A comprehensive examination of the existing scientific evidence on this phenomenon reveals a growing pattern over the last two decades, with the peak in scientific publications occurring between 2010 and 2020. This period of growth coincides with advances in early diagnosis and the implementation of neonatal screening programs and treatments, which arouse scientific interest. Despite this significant growth, since 2020 there has been a decrease in publications, which may be due to the need for all efforts to be directed toward studying the SARS-CoV-2 virus and its effects, treatments, and vaccines. This has led to a reduction in funding and in the priority of research related to other chronic diseases, including SCA. Furthermore, after years of intense research into neonatal screening, the treatment of vaso-occlusive crises and the use of hydroxyurea, some specific areas of SCA in pediatrics may have reached a level of knowledge saturation. Hence, the most recent studies address less explored aspects of the disease, such as the long-term effects of new genetic therapies, which are still at an early stage of research and have fewer publications.

The results of this rapid review suggest that the role of nurses in children with SCA is structural, within the context of multidisciplinary work. Nurses’ interventions are multifaceted and focus on the child and family, the team, and the health system. These interventions are not only important to address immediate problems in this population, but also to prevent long-term complications.

Family involvement is essential for proper SCA care, as family members play a key role in the early identification of complications and the implementation of preventive measures [48]. This aspect is particularly complex and benefits from an advanced nursing approach that combines scientific evidence with the epistemological structuring of knowledge. Consequently, analyzing the results in light of a theory that guides nursing practice can help bring the results of this review closer to nursing practice. Roy’s adaptation model is an epistemological approach that can be applied to the experience of SCA in a pediatric context, seeking to frame and explain the adaptive responses of children and their families, considering the physical, psychological, and social stress factors associated with hospitalization and dealing with the disease. According to Roy, people are adaptive systems that continually interact with the environment and are challenged by stimuli that affect their adaptive capacities [16]. For children with SCA, these triggers include intense pain due to vaso-occlusive crises, the need for frequent interventions, and emotional stress resulting from hospitalizations. In turn, families need to cope with the emotional impact of a child in pain, and manage the demands of ongoing care and the social and financial implications of chronic illness. This context should motivate the health system to create accessible clinical responses for the prevention of mental disorders in children and their families, mainly concerning depression and low self-esteem, which are not usually available for all SCA pediatric patients although this has long been a known necessity [49,50].

The literature highlights the importance of training to empower families in the prevention of vaso-occlusive crises [41]. Studies carried out in Latin America and Africa also suggest that educational interventions adapted to the cultural and socioeconomic context have the potential to improve adherence to treatment and reduce morbidity [40,51]. However, the lack of structured programs and longitudinal follow-up limits the effectiveness of these interventions, especially in developing countries where access to healthcare is restricted. Technology-based and nurse-led solutions, designed in a multiprofessional context, to provide training and empowerment to children and their families can be a solution to reduce accessibility problems in these contexts, although these tools are not equally available across all countries.

Symptom management in children with SCA is an area that needs special focus to improve quality-of-life and reduce complications from the disease [30,32,33,34,35,39,41,43,45,46]. Vaso-occlusive crises, characterized by intense and unpredictable pain, are the most common symptoms and require a rapid response to avoid complications and an increase in the child’s suffering [52]. The use of patient-controlled analgesia (PCA) seems to provide more effective pain relief and improves the hospitalization experience [34]. Other studies, such as the one developed by Donado et al. (2023), agree on the potential effectiveness of PCA for these children, especially when a continuous infusion is set on the device [53]. However, given the specificity of the medications used in these devices, challenges may arise from their application, whether regarding the setting (e.g., hospital at home) or the associated costs (e.g., in resource-limited countries).

For effective symptom management, educational interventions for families and children play a key role. Pain in children with SCA is often undervalued, partly due to the caregivers’ lack of knowledge about the severity of symptoms and treatment options [48]. Therefore, educating and training caregivers is crucial for early recognition of the signs of a crises and for implementing preventive measures. The family should be instructed on the importance of maintaining the child’s hydration and nutrition, preventing vaso-occlusive crises [54]. Demonstrating to children that they can promote their own care and empowering them can increase their sense of responsibility and confidence in themselves. Therefore, training should be carried out in such a way as to stimulate the autonomy of children and families so that they are empowered to make effective choices [55]. Sometimes, this is not possible given the excessive level of complexity of the materials which hinders their understandability. It is urgent that children assume their voice as co-creators of this content, facilitating its implementation to other pediatric SCA patients, which is in line with the recommendations for the delivery of children- and family-centered healthcare [56,57].

Symptom management in children with SCA must therefore be holistic and integrated, addressing not only the immediate resolution of pain but also prevention of it, and continuing education to improve adaptability and reduce associated complications. Health professionals must be attentive and prepared to understand gaps in knowledge and intervene with health education, generating greater adherence to self-care [58]. Concerning this need, community-based interventions are especially important to guarantee effective follow-up of these patients with family involvement [49]. Although this is an old perception, it remains a very contemporaneous problem given several challenges in the integration of care between hospital-based and primary care-based healthcare, as well as the connection with other families experiencing similar clinical situations with whom mutual-aid relationships could be established [59,60].

Early intervention is also essential to mitigate neurodevelopmental limitations. Ergo, early diagnosis through neonatal screening and laboratory tests is very important for identifying abnormal hemoglobin profiles and guiding the analysis of the spatial relationships of newborns with SCA, which can support the anticipation of health measures [7,61].

The nurse’s role also has a team focus since multidisciplinary intervention and professional collaboration are crucial in the treatment of SCA children, given the complexity of the condition, which involves multiple body systems and requires a broad approach. Collaboration with doctors, psychologists, social workers, and other professionals allows for more coordinated, individualized, and effective care planning, creating integrated interventions tailored to each individual’s needs and reducing hospitalization rates [35,48,62,63]. Older studies, such as that by Ballas et al. [62], have already suggested the importance of integrated and continuous management, showing that regular monitoring can prevent more serious complications, such as acute chest syndrome and stroke. Similarly, the study by Houwing et al. [35] reinforced that a collaborative approach between different specialties not only improves access to care but also provides a faster and more effective response to complications, reducing gaps in treatment. In this context, multidisciplinary training has been increasingly mentioned as a relevant topic for improving care for this population. According to Teixeira et al. [51], the continuous training of health professionals in hemoglobinopathies contributes to early diagnosis, reducing the risk of long-term complications.

Campelo et al. [48] highlighted the importance of nurses in providing education and emotional support to families, complementing medical treatment. An integrated approach contributes to adherence to treatment and improved quality of life for patients, especially when educational programs are used that promote self-care and understanding of the disease [64]. Continuous collaboration between different health professionals in this area is also highlighted as a tool that promotes access to care and allows for a quicker and more effective response to the complications of the disease, providing more personalized care tailored to the individual needs of patients [35,41,65]. In the context of therapeutic education, accessibility and effective communication between health professionals and families are essential components in guaranteeing quality care. Clear and accessible communication is a fundamental pillar in promoting adherence to therapeutic regimens, particularly in contexts of low health literacy and language barriers [33,34,66]. The promotion of communication strategies, such as the use of simple language, visual educational materials, and the involvement of interpreters, can significantly improve families’ understanding of care.

A partnership between professionals, children, and their families is essential to optimize communication and care. Building relationships of trust between them is essential for effective treatment. Developing intervention models with the designation of specific healthcare professionals to follow the trajectory of the disease, with deep knowledge about the specificity of a child’s therapeutic process, inherent family experiences, and the available support resources, may be a strategy with potential benefits for this population. This is a methodology already used in other complex chronic conditions, with positive outcomes for children and their family such as better integration of care, self-efficacy, parental role, and work–life balance [67,68,69].

Finally, the role of the child’s advocate is essential to ensure that children with SCA have access to appropriate care and that the necessary resources are available to carry out treatment. The defense of patients’ rights involves not only the promotion of better clinical practices but also the strengthening of public policies that ensure the inclusion of SCA on sociopolitical agendas as a priority condition for the health system [70]. The integration of these practices not only improves the quality of care provided but also contributes to equity in access to care and the optimization of disease management, which is particularly necessary in regions where SCA has a relatively high prevalence [71].

International studies point to the importance of neonatal screening programs and systematic vaccination as preventive measures that have significantly reduced morbidity and mortality in developed countries [44,51]. However, in developing countries, barriers to access, such as the lack of adequate infrastructure, insufficient funding, and low health literacy, are colossal challenges [35]. Advocacy by health professionals and nongovernmental organizations has been key to promoting the implementation of health policies aimed at improving accessibility to care and ensuring the use of effective therapies [35,38,40].

Despite these advances, there are still challenges in applying these recommendations in practice, especially in regions with limited resources. The literature highlights newborn screening, early education about SCA, vaccinations, safe transfusions, availability of hydroxyurea, antimicrobial prophylaxis, and the occurrence of complications (such as depression and strokes) as the main challenges in these settings, which may hinder the implementation of nursing interventions, mainly linked to early development, the use of technology and symptom control [20,72,73]. The evidence mentions some strategies to obviate the impact of these difficulties, mainly regarding policies to foster strong collaboration between high- and middle-income countries and low-resource settings, as well as partnerships between public and private sectors. These strategies may target early screening, but also proper planning of supportive care to ensure long-term sustainability, which should include symptom management, advanced treatment, mental health support, and the development of locally relevant educational materials for children and families [72,73,74]. To this aim, children, families, nurses, other health professionals, and stakeholders shall be included in policy development, in order for these policies to reflect real-world needs and promote efficient answers. The establishment of international networks to advocate for this collaboration may be a helpful strategy to increase health gains in resource-limited settings. The development of policies regarding the use of digital health tools should also be considered in these cases, given their potential to drive international collaboration and to generate better healthcare outcomes in these settings.

In a nutshell, nurses shall perform a unique role as the encounter between the children and family, the team, and the health system. With a clear focus in advocating for children with SCA and their family, nursing interventions with this population should be implemented through effective symptom management, adjusted training, and empowerment for therapeutic management and complication prevention, valuing multidisciplinary collaboration and fostering innovation answering to these needs.

Regarding the limitations of this rapid review, although it proves to be a useful tool for synthesizing evidence, it should be noted that, owing to the short time allocated to conducting the search, the short time frame, and the use of only three databases, there is a high risk of omitting relevant studies. In this type of study, the simplification of the search process and the inclusion of a few databases is a limitation as it does not cover all the studies on the subject under analysis. To obtain results with less risk of bias and greater robustness, a systematic review of the literature should be considered in the future. The fact that an expression with the PCC (population, concept, context) format was used reveals an adaptation of the Cochrane guidelines for conducting this type of review, which is also a limitation, since the guideline follows the PICO/PICo format. Although there was no screening criteria related to geographical location, the search revealed an over-representation of developed countries, which may hinder some difficulties in translating this knowledge into clinical practice in resource-limited settings. Another limitation is that there are no primary experimental studies, so it is necessary to further invest in this type of evidence to increase its degree of confidence.

## 5. Conclusions

Intervening with children with SCA and their families in a hospital setting requires a multidisciplinary approach focused on the child, the family, the team, and health policies to improve the care provided. The nursing role in this process in undeniably important.

Training children and their families through health education is essential to improve adherence to treatment, especially in the treatment of vaso-occlusive crises. Methods such as patient-controlled analgesia (PCA) stand out as effective tools for controlling pain and reducing complications. In addition, clear and accessible communication between health professionals and families is crucial to strengthening the bond, creating a good therapeutic relationship and consequently leading to greater adherence to the therapeutic regimen.

In short, the role of nurses in caring for children with SCA and their families is crucial to improving their quality of life, advocating for their needs and promoting interventions focused on partnership to reach symptom control, effective empowerment, promotion of therapeutic adherence, and adequate follow-up for a decreased rate of complications. Continuous training of nurses, at an undergraduate and a postgraduate level, in hemoglobinopathies and children with complex chronic diseases, but also regarding interprofessional work, is essential for the provision of high-quality nursing care. The inclusion of nurses in the development of public policies in this field shall be fostered by national and international nurses’ organizations given the value of their perspective to positively impact the SCA agenda in pediatric settings. Disparities between high-income countries and resource-limited settings highlight the need for inclusive public policies, collaboration between these contexts, and programs to strengthen health services, especially in the most affected regions, therefore contributing to sustainable development goals through universal health coverage.

Future directions, such as as the adjustment of clinical protocols based on scientific evidence, innovation in educational strategies, and multidisciplinary collaboration are essential for facing the challenges imposed by this disease. Although this review has provided relevant insights, we believe that more comprehensive systematic reviews and primary experimental studies focusing on the impact of nursing and multiprofessional interventions across the continuum of children’s socialization contexts are needed to strengthen the evidence, broaden the scope, and guide future healthcare practice and policy within this population.

## Figures and Tables

**Figure 1 healthcare-13-00413-f001:**
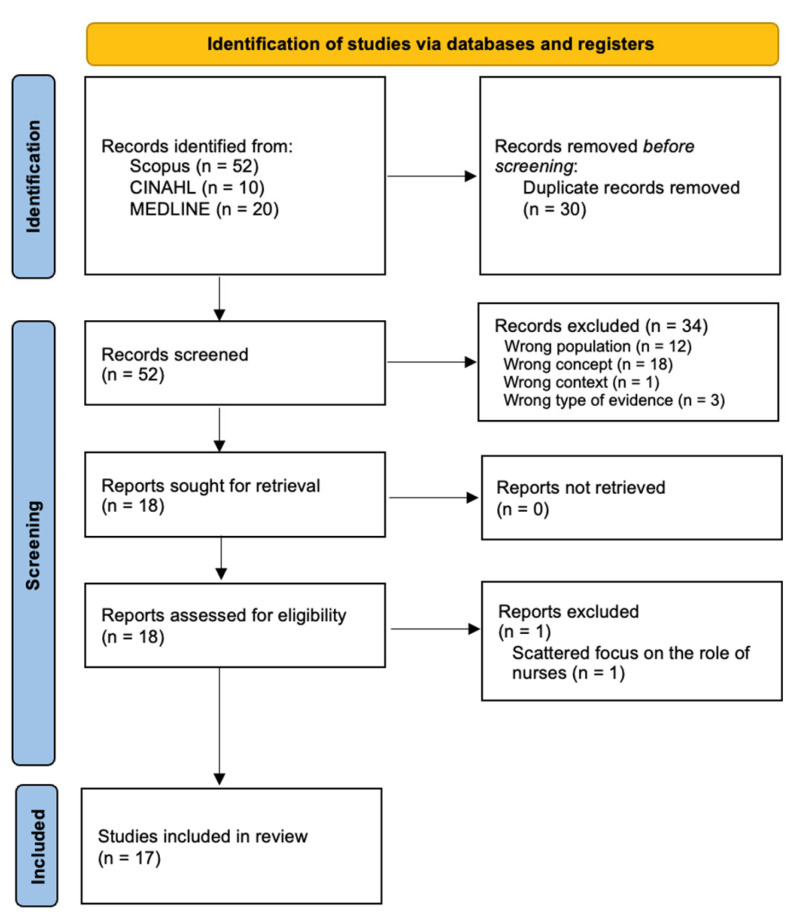
PRISMA flow diagram reporting the process of identification, screening, and inclusion of articles in this rapid review [24].

## Data Availability

No new data were created or analyzed in this study.

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
