# Peer review of "The Role of Nurses Caring for Children Diagnosed with Sickle Cell Anemia and Their Families in a Hospital Setting: A Rapid Review of the Recent Literature"

_healthcare, 2025, doi:10.3390/healthcare13040413_

Round 1

Reviewer 1 Report

Comments and Suggestions for Authors

It is considered a well-prepared work with a robust methodology, but I suggest some considerations to improve its bibliographic search:

- The use of a single database (Scopus) may have omitted relevant studies.

- The review did not include experimental primary studies, which limits the strength of evidence for some interventions.

- Although studies from different countries are included, there is an over-representation of studies from developed countries, which may limit applicability in resource-limited settings.

- Some aspects, such as specific interventions to prevent long-term complications, could have been explored in more detail.

- Barriers to implementing the identified nursing interventions could have been further explored, especially in resource-limited settings.

Author Response

Reviewer 1

1. It is considered a well-prepared work with a robust methodology, but I suggest some considerations to improve its bibliographic search.
R: The authors thank reviewer 1 for the constructive feedback and the will to improve our article. Serious and rigorous work has been done to meet your suggestions, as the authors agree on their positive impact on the quality of the manuscript.

2. The use of a single database (Scopus) may have omitted relevant studies.
R: Thank you for this comment. The authors acknowledged this matter as a strong limitation of the article and, in line with recommendations from most reviewers, conducted an updated search in Scopus and a new search in CINAHL and MEDLINE databases, with a larger time frame to include studies published in English between 2019-2024 (last 5 years). This search yielded 4 new studies, which have been added to our review.

3. The review did not include experimental primary studies, which limits the strength of evidence for some interventions.
R: Thank you for raising this important question. Experimental studies on children with sickle cell anemia would be really important to identify meaningful knowledge to answer the review question. Therefore, the authors did not exclude this type of studies from the scope of articles included in the review. The fact that experimental studies were not included originated in the absence of identification of these studies in the three databases that were searched by the authors. Given that the authors agree that this fact can limit the strength of evidence for some interventions, this matter is mentioned as a limitation.

4. Although studies from different countries are included, there is an over-representation of studies from developed countries, which may limit applicability in resource-limited settings.
R: Thank you for your interesting comment. Although there were no criteria based on geographical location to tailor the selection process, the authors recognized that there is an over-representation of studies from developed countries, probably because those are the ones where more funding is available to conduct this kind of research. This matter is now mentioned in limitations given that it may be a factor introducing bias in results and hinder some difficulties translating this knowledge into resource-limited settings. 

5. Some aspects, such as specific interventions to prevent long-term complications, could have been explored in more detail. 

R: Thank you for your suggestion. Nursing interventions related to this population contribute collectively to prevent long-term complications, since none of them alone can ensure a long lasting effect. This matter was outlined on page 18. Therefore, and in light of the suggestions of other reviewers, several interventions were detailed further and express the potential to prevent long-term complications, such as clinical responses for mental health disorders prevention, involvement of children in the production of clinical leaflets to improve their understandability, community follow-up and the development of specific care pathways to enhance the accessibility to healthcare. Please refer to the discussion section for further development. 

6. Barriers to implementing the identified nursing interventions could have been further explored, especially in resource-limited settings.

R: Thank you for your suggestions. Indeed, there are a lot of barriers for the identified nursing interventions, which may not always be fully applicable depending on context. In line with the recommendation of most reviewers, barriers regarding resource-limited settings were explored further in discussion, along with some evidence supporting such a perspective.

Reviewer 2 Report

Comments and Suggestions for Authors

Dear Authors,

The introduction does a good job of presenting the context of sickle cell anemia (SCA), its prevalence and impact, especially in children. However, it would be beneficial to include more quantitative data to reinforce the overall magnitude of the problem. The research question is well defined, but could be more clearly explained as a direct objective in the introduction.

The PRISMA methodology was well applied. However, the justification for using only the Scopus database is limited. The inclusion of more relevant databases, such as MEDLINE or CINAHL, is recommended to strengthen the robustness of the findings. The absence of a formal assessment of the quality of the studies limits the credibility of the results presented. An assessment approach, even a rapid one, would have added greater value.

The results are well organized into relevant categories (children and family, team and health system). The presentation of the data in tables and visual diagrams is clear and effective. However, some statements lack detail or practical examples, such as the description of specific interventions in symptom management.

The discussion contextualizes the findings based on different global scenarios and highlights the importance of adaptive cultural and economic strategies. Although relevant, there was a lack of depth on how these strategies can be implemented, especially in low-income countries.

The conclusion summarizes the findings well and emphasizes the importance of inclusive policies and the central role of nurses. It could include more detailed practical recommendations for implementing best practices in pediatric nursing.

Author Response

Reviewer 2

1. The introduction does a good job of presenting the context of sickle cell anemia (SCA), its prevalence and impact, especially in children. However, it would be beneficial to include more quantitative data to reinforce the overall magnitude of the problem.

R: I appreciate the feedback. To reinforce the magnitude of the problem, we added quantitative data on the global incidence of sickle cell anemia (SCA), highlighting that approximately 300,000 to 400,000 children are born annually with the disease, with greater prevalence in Sub-Saharan Africa, the Indian subcontinent and some regions of the Middle East and Latin America. Additionally, the authors emphasized the infant mortality rate in low-income countries, where up to 90% of children with SCA can die before the age of five without treatment adequate. These additions aim to strengthen the introduction and provide a more concrete overview of the relevance of the disease. These changes were made on page 2. If you have any further suggestions, we remain at your disposal for further adjustments.

2. The research question is well defined, but could be more clearly explained as a direct objective in the introduction.

R: Thank you for your suggestion. This article aims to provide a rapid but thorough understanding of the nursing role caring for pediatric patients with sickle cell anemia and their families, which is important given that its conclusions can inform clinical practice and inspire the development of new studies in the field. This is especially relevant due to the increasing complexity of this disease in parallel with healthcare accessibility difficulties for this population. Such aspects were clarified in the background section.

3. The PRISMA methodology was well applied. However, the justification for using only the Scopus database is limited. The inclusion of more relevant databases, such as MEDLINE or CINAHL, is recommended to strengthen the robustness of the findings. 

R: Thank you for this comment. The authors acknowledged this matter as a strong limitation of the article and, in line with recommendations from most reviewers, conducted an updated search in Scopus and a new search in CINAHL and MEDLINE databases, with a larger time frame to include studies published in English between 2019-2024 (last 5 years). This search yielded 4 new studies, which have been added to our review.

4. The absence of a formal assessment of the quality of the studies limits the credibility of the results presented. An assessment approach, even a rapid one, would have added greater value.

R: Thank you for raising this point. In line with the recommendation of some reviewers, and given that the quality assessment of studies included in rapid reviews is recommended by Cochrane, the authors have now revised this part of the manuscript using a quality appraisal process. The grade used by Hawker et al. (2002) was used to assess the quality of primary studies, while JBI recommendations were used to assess the quality of systematic and narrative reviews. This matter was explained in the article and the conclusion of this evaluation was mentioned in the results. Please refer to Supplementary File A for details on this process.

5. The results are well organized into relevant categories (children and family, team and health system). The presentation of the data in tables and visual diagrams is clear and effective. However, some statements lack detail or practical examples, such as the description of specific interventions in symptom management.

R: Thank you for your input. The authors have revised the nursing role in symptom management in order to provide a more representative summary of the findings. Please refer to page 15.

6. The discussion contextualizes the findings based on different global scenarios and highlights the importance of adaptive cultural and economic strategies. Although relevant, there was a lack of depth on how these strategies can be implemented, especially in low-income countries.

R: Thank you for your suggestions. In line with the recommendation of most reviewers, barriers regarding resource-limited settings were explored further in discussion, along with some evidence supporting such a perspective. Also, the section on the role of nursing on the implementation of the described strategies was amplified and summarized in the end of the discussion, before limitations.

7. The conclusion summarizes the findings well and emphasizes the importance of inclusive policies and the central role of nurses. It could include more detailed practical recommendations for implementing best practices in pediatric nursing.

R: Thank you for your valuable feedback. The conclusion was revised to include such practical recommendations. The nursing role should be implemented through interventions focused in partnership to reach symptom control, effective empowerment, promotion of therapeutic adherence and adequate follow-up for a decreased rate of complications.

Reviewer 3 Report

Comments and Suggestions for Authors

Reviewer Comments: Lack of Originality in the Research Focus: The study appears to address an important topic, but the focus on nurses' roles in managing sickle cell anemia (SCA) has been extensively explored in existing literature. The review does not seem to provide a novel perspective or substantially new insights, making its contribution to the field less significant.

 Methodological Concerns: The title emphasizes the phrase "I have the right to be pain-free," but the abstract lacks detailed information on how this aspect (pain management advocacy or rights-based perspective) was systematically reviewed or analyzed. The methodology described does not align with the title's emphasis, which could lead to misinterpretation or unmet reader expectations.

Limited Scope of Analysis: Although the rapid review identifies nurses' structural and multidisciplinary roles, it does not adequately highlight innovative or transformative interventions that could advance clinical practice. A broader exploration of unique nursing strategies, particularly in resource-limited settings, would enhance its practical value.

Insufficient Focus on Practical Implications: While the abstract mentions the need for greater investment, it does not articulate specific recommendations or actionable strategies for improving nursing care or system-level interventions. This limits the study’s applicability to real-world nursing practice and policy-making.

Overemphasis on Recent Literature: The study's restriction to articles from 2021-2024 narrows the scope unnecessarily, potentially comitting foundational studies that could provide a more comprehensive understanding of the topic. This limits the depth and historical context of the analysis.

Ambiguity in the Title: The phrase "I have the right to be pain-free" is compelling but does not reflect the study's primary aim or findings as described in the abstract. It may set unrealistic expectations for readers about the review’s depth on pain management advocacy or legal frameworks.

Recommendation: Based on these observations, I recommend declining the manuscript in its current form. However, I encourage the authors to: Refocus the review to address a novel or underexplored aspect of nursing care for children with SCA. Expand the scope of the literature review to incorporate a broader timeframe or diverse healthcare contexts. Reframe the title to align more closely with the study's methodology and findings. Enhance the practical implications by providing clear, actionable recommendations for nursing practice, training, or healthcare policy.

Comments on the Quality of English Language

NA

Author Response

Reviewer 3

1. Methodological Concerns: The title emphasizes the phrase "I have the right to be pain-free," but the abstract lacks detailed information on how this aspect (pain management advocacy or rights-based perspective) was systematically reviewed or analyzed. The methodology described does not align with the title's emphasis, which could lead to misinterpretation or unmet reader expectations.
Ambiguity in the Title: The phrase "I have the right to be pain-free" is compelling but does not reflect the study's primary aim or findings as described in the abstract. It may set unrealistic expectations for readers about the review’s depth on pain management advocacy or legal frameworks.

R: Thank you for this important feedback. The authors understand your concerns regarding the title. The intention was not to induce any unrealistic expectations for readers but to highlight the importance of pain control as a significant health outcome for this population. However, we have revised the title trying to optimize its alignment with the manuscript: “The role of nurses caring for children diagnosed with sickle cell anemia and their families in a hospital setting: a rapid review of recent literature.”

2. The English could be improved to more clearly express the research.

R: Thank you for your suggestion. The text was proofread by the authors and revised for grammatical errors and unclear phrasing. After this process, the manuscript was submitted to a native speaker for verification. We hope that the quality of text regarding language is now better than the initial version.

3. Lack of Originality in the Research Focus: The study appears to address an important topic, but the focus on nurses' roles in managing sickle cell anemia (SCA) has been extensively explored in existing literature. The review does not seem to provide a novel perspective or substantially new insights, making its contribution to the field less significant.

R: Thank you for your perspective. While the authors recognize that sickle cell anemia is a topic in which there is a high amount of evidence available, nursing intervention regarding this matter especially in pediatric patients has not been explored with the same detail as in the adult population. Regarding the increasing prevalence of pediatric-onset of sickle cell anemia globally, it is important to understand the role of nursing in these cases. Given that the authors did not find a review specifically about this topic, this review is an important addition to the literature in the perspective of the authors, opening further opportunities of research in the field to better understand this topic, such as experimental studies and systematic reviews. The background was revised to better express the novelty of this research.

4. Overemphasis on Recent Literature: The study's restriction to articles from 2021-2024 narrows the scope unnecessarily, potentially comitting foundational studies that could provide a more comprehensive understanding of the topic. This limits the depth and historical context of the analysis.

R: Thank you for your perspective. While the authors understand that a date restriction may limit the scope of the evidence and potentially omit foundational studies on the topic, our goal is to identify recent evidence that can be evaluated for potential integration into clinical practice. Because of this intention, the evidence to be considered should be desirably novel and recent. However, recognizing the specially strict time frame proposed, the authors have conducted a new search, as a result of the recommendations from most reviewers, with a larger time frame to include studies published in English between 2019-2024 (last 5 years). This search yielded 4 new studies, which have been added to our review. Still, this matter will be added to the limitations of the study. As the authors mentioned in conclusions, more comprehensive systematic reviews are still needed to strengthen the evidence and guide future health practices and policies.

5. Limited Scope of Analysis: Although the rapid review identifies nurses' structural and multidisciplinary roles, it does not adequately highlight innovative or transformative interventions that could advance clinical practice. A broader exploration of unique nursing strategies, particularly in resource-limited settings, would enhance its practical value.

R: Thank you for your suggestions. In line with the recommendation of most reviewers, barriers regarding resource-limited settings were explored further in discussion, along with some evidence supporting such a perspective. 

6. Insufficient Focus on Practical Implications: While the abstract mentions the need for greater investment, it does not articulate specific recommendations or actionable strategies for improving nursing care or system-level interventions. This limits the study’s applicability to real-world nursing practice and policy-making.
R: Thank you for your opinion and expertise on the subject. Nurses shall perform a unique role as the encounter between the children and family, the team and the health system. With a clear focus in advocating for children with SCA and their family, nursing interventions with this population should be implemented through effective strategies, which are now described in the discussion section. 

7. Based on these observations, I recommend declining the manuscript in its current form. However, I encourage the authors to: Refocus the review to address a novel or underexplored aspect of nursing care for children with SCA. Expand the scope of the literature review to incorporate a broader timeframe or diverse healthcare contexts. Reframe the title to align more closely with the study's methodology and findings. Enhance the practical implications by providing clear, actionable recommendations for nursing practice, training, or healthcare policy.

R: Thank you for your comment. The authors have considered your feedback and adjusted the manuscript accordingly. We would like to thank you for your constructive points and hope the revisions made are now in line with your quality standards.

Reviewer 4 Report

Comments and Suggestions for Authors
  • Introduction: Several of the researcher's claims appear to lack objective evidence. For instance, many of the statements between lines 59 and 74 are presented without supporting references. While not every sentence requires a citation, please provide references to substantiate key claims within this section.

  • Purpose of Figure 1: What is the intended purpose of including <Figure 1>? Please clarify its role in the research. Currently, its inclusion seems to disrupt the flow and coherence of the discussion.

  • Details in Table 1: When summarizing the research characteristics in <Table 1>, it would be helpful to provide more detailed explanations of the subject characteristics, research design, variables, and results. This additional context would improve the table’s clarity and usefulness.

  • Supplementary Table for Results: In addition to <Table 1>, consider creating a separate table to detail the studies mentioned in the results section. For example, you could construct a table with the studies listed along the vertical axis and the titles of the results along the horizontal axis. This format would enable readers to quickly identify which elements are included in each study.

Author Response

Reviewer 4

1. Introduction: Several of the researcher's claims appear to lack objective evidence. For instance, many of the statements between lines 59 and 74 are presented without supporting references. While not every sentence requires a citation, please provide references to substantiate key claims within this section.
R: Thank you for your comment. The authors have considered your feedback and adjusted the manuscript accordingly. More references were added, including the statement you mentioned. Please refer to the introduction for further details.

2. Purpose of Figure 1: What is the intended purpose of including <Figure 1>? Please clarify its role in the research. Currently, its inclusion seems to disrupt the flow and coherence of the discussion.

R: Thank you for your suggestion. Image 1 was removed following your suggestion to respect the flow of the discussion.

3. Details in Table 1: When summarizing the research characteristics in <Table 1>, it would be helpful to provide more detailed explanations of the subject characteristics, research design, variables, and results. This additional context would improve the table’s clarity and usefulness.

R: Thank you for your feedback. Table 1 was revised to provide more information following your suggestion. 

4. Supplementary Table for Results: In addition to <Table 1>, consider creating a separate table to detail the studies mentioned in the results section. For example, you could construct a table with the studies listed along the vertical axis and the titles of the results along the horizontal axis. This format would enable readers to quickly identify which elements are included in each study.
R: Thanks for the suggestion. In order not to overload the number of elements apart from the text, we added this information in Image 2, which presents the categories in a schematic illustration.

Reviewer 5 Report

Comments and Suggestions for Authors

Thank you for sharing: 

1- What is "FCA" in the abstract? please try to avoid abbreviation in abstract or at least put what is referred to.

2-The introduction lacks a strong link to the importance or relevance of the nursing role. Please elaborate more about this. 

3-The purpose of the abstract and introduction is very general.  Please be specific (to pain and/or other symptoms) 

4-Please identify a clear research gap, significance, and the novelty of this review in the introduction.

5- Regarding the use of Scopus only in the second phase. The justification is not clear, and this negatively affects the comprehensive review. 

6- I do not see a quality assessment of the included studies. 

7-Please explain more about PCC and PICO formats and difference between them in the methods

8-Did you do pilot testing or inter-reviewer reliability? If yes, please explain, and if no, how you minimise the bias and inconsistency,

9-The results are fragmented and lack coherence, with excessive reliance on general statements rather than highlighting critical insights.

10-categories (child and family, team, health system) is not well-supported by detailed examples from the studies. I recommend adding new columns to classify which themes are covered for each study in Table 1. Summary of selected articles.

11-Limitations should include the use of scopus database, absence of quality assessment, pilot testing or inter-reviewer reliability.

12-Recommendation for the experimental study is needed to be added 

13-the discussion needs to be critically analyzing the implications of the findings or comparing them with existing literature

Author Response

Reviewer 5

1. What is "FCA" in the abstract? please try to avoid abbreviation in abstract or at least put what is referred to

R: Thank you for your question. This was a mistake, for which the authors apologize. It is now corrected to SCA - Sickle Cell Anemia.

2. The introduction lacks a strong link to the importance or relevance of the nursing role. Please elaborate more about this.
R: Thank you for your important note. Sickle cell anemia is a chronic complex condition demanding consistent and individualized interventions. Talking about children with this disease requires healthcare professionals to think about the multidimensional impact of SCA in childhood. This impact overcomes disease-related issues and expands itself to child development challenges. Nurses are in the front row of healthcare, serving as a connection point between specialized healthcare providers, children and families. Their action, whether hospital or community-based, can significantly increase health gains for children with SCA and their families. Therefore, a literature review on the nursing role for this population is highly relevant and can provide new insights for quality improvement and patient safety. These aspects were better articulated in the background section.

3. The purpose of the abstract and introduction is very general.  Please be specific (to pain and/or other symptoms).
R: Thank you for your feedback. The introduction aims to explain the state of the art regarding this topic. According to the suggestion of other reviewers, several information was added to better explore the importance of this review and provide a more clear purpose of the conducted research. The authors are open to new suggestions.

4. Please identify a clear research gap, significance, and the novelty of this review in the introduction.

R: Thank you for your suggestion. This review is an important step to identify relevant material for evidence-based practice, helping to enhance nursing care to this population. To the best of the authors' knowledge, this is the first literature review focusing specifically on the nursing role in children with SCA. This consideration and other topics supporting the research gap were added to the background section.

5. Regarding the use of Scopus only in the second phase. The justification is not clear, and this negatively affects the comprehensive review. 

R: Thank you for this comment. The authors acknowledged this matter as a strong limitation of the article and, in line with recommendations from most reviewers, conducted an updated search in Scopus and a new search in CINAHL and MEDLINE databases, with a larger time frame to include studies published in English between 2019-2024 (last 5 years). This search yielded 4 new studies, which have been added to our review.

6. I do not see a quality assessment of the included studies. 

R: Thank you for raising this point. In line with the recommendation of some reviewers, and given that the quality assessment of studies included in rapid reviews is recommended by Cochrane, the authors have now revised this part of the manuscript using a quality appraisal process. The grade used by Hawker et al. (2002) was used to assess the quality of primary studies, while JBI recommendations were used to assess the quality of systematic and narrative reviews. This matter was explained in the article and the conclusion of this evaluation was mentioned in the results. Please refer to Supplementary File A for details on this process.

7. Please explain more about PCC and PICO formats and difference between them in the methods.

R: Thank you for this suggestion. The PCC framework aims to structure the research question in population, concept and context, which is suitable for this rapid review given that the aim is to identify recent evidence on the role of nurses in interventions with children with SCA and their families in a hospital setting, The PICO framework is more suitable for reviews were there is a need to identify the outcome of an intervention comparing to other factors in a certain population. Since this rapid review does not intend to introduce comparators or outcomes a priori, the PCC framework was the chosen approach for constructing the research question. This information was added to the methods’ section.

8. Did you do pilot testing or inter-reviewer reliability? If yes, please explain, and if no, how you minimise the bias and inconsistency.

R: Thank you for your question. Pilot testing was not performed since two reviewers independently screened all manuscripts for inclusion and data extraction. An inter-reviewer reliability score was missing in the first version of the manuscript and it is now added. Please refer to section 2.4. Data extraction.

9. The results are fragmented and lack coherence, with excessive reliance on general statements rather than highlighting critical insights.
R: The authors appreciate the feedback. To improve the coherence and flow of the results, the text has been revised to make it more articulated and logical. We have put significant effort in reducing fragmentation by reorganizing the information in a more connected manner, ensuring that each point contributes to a clearer and more in-depth analysis.

10. Categories (child and family, team, health system) is not well-supported by detailed examples from the studies. I recommend adding new columns to classify which themes are covered for each study in Table 1. Summary of selected articles.

R:Thanks for the suggestion. As mentioned in previous comments of other reviewers, and in order not to overload the number of elements apart from the text, we added this information in Image 2, which presents the categories in a schematic illustration.

11. The discussion needs to be critically analyzing the implications of the findings or comparing them with existing literature.

R: Thank you for your suggestion. The discussion was optimized regarding the implications of findings and comparison with existing literature, mainly regarding family support, community follow-up, training methodologies and reference professionals. 

12. Recommendation for the experimental study is needed to be added.

R: Thank you for your suggestion. The authors agree with this need as it is very important to increase confidence in study findings. This recommendation was optimized in the discussion section and it was also placed in the conclusion section.

13. Limitations should include the use of scopus database, absence of quality assessment, pilot testing or inter-reviewer reliability.

R: Thank you for your comment. Since these matters were addressed by the authors, they were not included in limitations. However, the limitations section was revised according to inputs from other reviewers. The authors are available to reframe it if you think it is necessary.

Round 2

Reviewer 3 Report

Comments and Suggestions for Authors

1. Title & Abstract:

-The title is clear

--The abstract is organized but could be enhanced by explicitly indicating the number of studies examined, the principal findings, and the significant gaps in the literature.

-Clarify the methodology used for selecting studies (i.e., number of databases searched, inclusion/exclusion criteria).

-The phrase "right to be pain-free" in the conclusion is impactful but might require elaboration in the discussion section.

2. Introduction:

-The introduction offers a solid background on SCA but requires improved incorporation of the nursing perspective at the outset.

-Include more details on why a rapid review was chosen instead of a systematic review or scoping review.

- Some sentences are redundant, such as describing the global prevalence of SCA in multiple sections.

3. Methods: The search strategy needs more details on:

-Which databases were searched? (It mentions Scopus, CINAHL, and MEDLINE, but were others considered?)

-Search string or terms used? (Explicit mention of keywords used in Boolean searches would improve transparency.)

-Why was only Scopus used in the final phase? This decision contradicts Cochrane guidelines and needs justification.

-Clarify the quality appraisal process. The study mentions Hawker et al. (2002) for quality assessment, but also JBI recommendations. Were both used?

-The exclusion criteria for studies could be further detailed—especially why some were removed (e.g., studies without nurse involvement).

4. Results

-The PRISMA flowchart is well-represented, but the number of initial search results vs. included studies should be explained more clearly.

-The summary table (Table 1) is informative, but consider:

Including a column for study limitations.

-Standardizing study designs (e.g., "Primary Study – Quantitative" should be formatted consistently).

-Some numerical inconsistencies exist, e.g.:

"Among the 37 articles identified, 13 were included" (later, different numbers are mentioned).

-Some studies are described as covering 2023–2024, while others mention 2021–2024.

5. Discussion

-The discussion effectively highlights the multidimensional role of nurses, but some subsections repeat information from the results (e.g., symptom management, family empowerment).

-The comparison between high-income and low-resource settings is important, but more emphasis on policy recommendations would be useful.

-The mention of Roy’s Adaptation Model is interesting but would be more effective if discussed earlier in the review rather than in the discussion.

-There is limited discussion on telemedicine and digital health interventions, which could be valuable for nurses in remote or resource-limited settings.

6. Conclusion

-The conclusion is generally strong but needs to:

Summarize the practical implications for nurses (e.g., training, policy recommendations).

-Avoid repeating information already discussed.

Suggest future research directions, especially for multidisciplinary interventions.

Author Response

1. The title is clear. The abstract is organized but could be enhanced by explicitly indicating the number of studies examined, the principal findings, and the significant gaps in the literature.

R: The authors would like to thank Reviewer 3 for the suggestion. The abstract was revised in order to comply with the indications you mentioned. 

2. Clarify the methodology used for selecting studies (i.e., number of databases searched, inclusion/exclusion criteria).

R: Thank you for the suggestion. This information was added to the abstract.

3. The phrase "right to be pain-free" in the conclusion is impactful but might require elaboration in the discussion section.

R: Thank you for pointing this out. This mention corresponded to a previous version of the manuscript, where this approach was taken by the authors. After careful consideration of previous comments from the reviewers, the authors decided to remove the mention to a rights-based approach to the topic. Therefore, the conclusion of the abstract is now aligned with the rest of the manuscript. 

4. The introduction offers a solid background on SCA but requires improved incorporation of the nursing perspective at the outset.

R: The authors thank you for the comment on the introduction. Indeed, the suggested changes have been duly considered. The goal was to reinforce the essential role of nursing from the outset, establishing a clearer connection between the clinical aspects of sickle cell anemia (SCA) and the implications for nursing care.

5. Include more details on why a rapid review was chosen instead of a systematic review or scoping review.

R: Thank you for your comment. A more detailed explanation about the option to conduct a rapid review was added to the introduction. This option is related to the shattered pace to which the incidence of SCA has been growing compared with the slow rhythm of scientific production on the topic. Especially regarding nursing interventions, this issue generates a need to quickly identify studies that may be transferred to clinical practice, contributing to evidence-based practice.

6. Some sentences are redundant, such as describing the global prevalence of SCA in multiple sections.

R: Thank you for your comment. The global prevalence of SCA is described in detail in the introduction. Further mentions of SCA prevalence in the text were assessed and redundant sentences were removed.

7. The mention of Roy’s Adaptation Model is interesting but would be more effective if discussed earlier in the review rather than in the discussion.

R: Thank you for the suggestion. We have revised the structure of the review and included Roy's Adaptation Model theory in the introduction. This adjustment improves the flow of the text and strengthens its relevance to the overall analysis. However, a mention of Roy’s Adaptation Model was also made in the discussion section, aiming to provide a more interpretative insight into the link between this theory and the findings from this review.

8. The search strategy needs more details on: Which databases were searched? (It mentions Scopus, CINAHL, and MEDLINE, but were others considered?)

R: Thank you for pointing this out. As mentioned in the manuscript, in the beginning of section 2.2., an exploratory search was carried out in databases appropriate for the subject under research (CINAHL, MEDLINE, MedicLatina, Scopus, Web of Science and Academic Search Complete) and a search expression was developed during this phase. It became clear that this area is a field of heuristic availability, and not all of the databases returned results. The definitive search itself was rerun on the Scopus, CINAHL and MEDLINE databases with the previously tested search expression with maximized specificity and sensitivity. The choice of these databases was linked to their multidimensional nature and to higher probability of returning results, given that these were the databases with more articles available in the previous exploratory search. Given that the authors consider the information presented in the manuscript to clearly answer the question, no further changes were made regarding this comment. We are happy to revisit this if necessary.

9. The search strategy needs more details on: Search string or terms used? (Explicit mention of keywords used in Boolean searches would improve transparency.)

R: Thank you for your comment. During the exploratory search phase, the authors have screened the most usual terms used in studies related to nursing interventions in children with sickle cell anemia, which made it possible to test a preliminary version of the search expression. This expression was then applied to the databases used for the exploratory search and refined according to the results. The search expression presented in the manuscript corresponds to the final version, which uses natural language terms given the higher number of results compared with indexed language terms. The description of the search expression is now highlighted in italic and in a sole paragraph to improve its readability and a more detailed explanation about the development process of this expression was added to the manuscript. Please refer to the last two paragraphs of section 2.2.

10. The search strategy needs more details on: Why was only Scopus used in the final phase? This decision contradicts Cochrane guidelines and needs justification.

R: Thank you for your question. As mentioned in previous comments, the mention of the use of Scopus as the only database in the final search belonged to a previous version of the manuscript and it is corrected in the manuscript. Scopus, CINAHL and MEDLINE databases were used in the final search, complying with Cochrane recommendations, and the results are described in the PRISMA flow diagram.

11. Clarify the quality appraisal process. The study mentions Hawker et al. (2002) for quality assessment, but also JBI recommendations. Were both used?

R: Thank you for your feedback. Both tools were used given the different nature of studies included. The Hawker et al. (2002) tool for quality appraisal only assesses primary studies. Given that the results included secondary studies, JBI recommendations were used to assess these reviews. A general compliance rate was calculated to express a generic quality of the included evidence. No studies were excluded based on the results of this appraisal process. The text was refined to describe this process more clearly.

12. The exclusion criteria for studies could be further detailed—especially why some were removed (e.g., studies without nurse involvement).

R: Thank you for your suggestion. Exclusion criteria were defined to increase the specificity of the review, given that only studies mentioning nursing interventions and focused on the nursing role concerning children with sickle cell anemia would contribute to answering the review question. Therefore, studies that addressed interventions without nurse involvement or without implications for nursing and that were exclusively conducted by other professionals, as well as interventions not directly related to the provision of direct care, were excluded. The decision of excluding one article by full text was made due to the fact that, although nurses were involved in the study, its focus was not their intervention and, therefore, it did not have a meaningful contribution to the present review. A detailed explanation about this decision was added to the manuscript, in the beginning of the results’ section. 

13. The PRISMA flowchart is well-represented, but the number of initial search results vs. included studies should be explained more clearly.

R: Thank you for your input. The text has been revised, and the necessary clarifications have been made to ensure better understanding and complying with the PRISMA flow diagram.

14. The summary table (Table 1) is informative, but consider: Including a column for study limitations.

R: Thank you for your suggestion. While the authors agree on including information about study limitations, given its importance to frame studies conclusions to answer the research question, we believe that the inclusion of a new column would disrupt information display in Table 1. Therefore, this information was added in the column of “Main findings” at the end of each paragraph.

15. The summary table (Table 1) is informative, but consider: Standardizing study designs (e.g., "Primary Study – Quantitative" should be formatted consistently).

R: Thank you for your feedback. The authors revised Table 1 in order to format methodology designs consistently. Therefore, two sections were created inside the column for the type of study (e.g. Primary study - Qualitative) and for designs or methods (e.g. Semi-structured interviews) according to the available information. We hope that the information is more clear now.

16. Some numerical inconsistencies exist, e.g.: "Among the 37 articles identified, 13 were included" (later, different numbers are mentioned).

R: Thank you for pointing this out. We have carefully revised the numerical data and corrected the inconsistencies. The numbers now align with the PRISMA flow diagram, ensuring accuracy and coherence throughout the text.

17. Some studies are described as covering 2023–2024, while others mention 2021–2024.

R: Thank you for your input. In fact, the authors highlight the recent nature of the included articles and there were two considerations about this topic with different timeframes within results and discussion. The text has been revised to ensure a more descriptive analysis and better coherence.

18. The discussion effectively highlights the multidimensional role of nurses, but some subsections repeat information from the results (e.g., symptom management, family empowerment).

R: Thank you for your comment. The discussion section was revised from top to bottom to identify repeated information and to improve coherence in the text, mainly regarding symptom management, family empowerment and advocacy. Please refer to this section to assess this adjustment. 

19. The comparison between high-income and low-resource settings is important, but more emphasis on policy recommendations would be useful.

R: Thank you for your feedback. The authors agree that policy recommendations are extremely important to foster the need to develop collaboration between high income and low-resource settings. Given that the authors had already invested some time developing suggestions for policy making regarding this topic, the text was revised and more recommendations were made. Please refer to the last paragraphs of discussion. 

20. There is limited discussion on telemedicine and digital health interventions, which could be valuable for nurses in remote or resource-limited settings.

R: Thank you for pointing this out. Digital-health interventions are indeed a potential way to reduce complications in resource-limited settings. Nurses can have a significant role in the development of such tools, whether they are directed for training and empowerment or for international collaboration. Two sentences regarding this aim were added to the discussion section. 

21. The conclusion is generally strong but needs to: Summarize the practical implications for nurses (e.g., training, policy recommendations).

R: Thank you for your suggestion. Implications for nurses were now summarized and clarified in the conclusion to increase their understandability and transfer to clinical practice.

22. Avoid repeating information already discussed.

R: Thank you for your recommendation. The text was revised in order to meet your suggestion, removing already discussed information.

23. Suggest future research directions, especially for multidisciplinary interventions.

R: Thank you for your suggestion. This topic was added to research implications for the future in the conclusion.

The authors would like to acknowledge the contribution of Reviewer 3 to increase the scientific merit of the manuscript. Thank you for your meaningful comments. We hope to benefit from your expertise in the future.

Reviewer 5 Report

Comments and Suggestions for Authors

Thank you for your excellent work. No comments.

Author Response

1. Thank you for your excellent work. No comments.

R: The authors would like to acknowledge the contribution of Reviewer 5 to increase the scientific merit of the manuscript. Thank you for your meaningful comments. We hope to benefit from your expertise in the future.